# MED16 Promotes Tumour Progression and Tamoxifen Sensitivity by Modulating Autophagy through the mTOR Signalling Pathway in ER-Positive Breast Cancer

**DOI:** 10.3390/life12101461

**Published:** 2022-09-20

**Authors:** Han Li, Kang Li, Dan Shu, Meiying Shen, Zhaofu Tan, Wenjie Zhang, Dongyao Pu, Wenhao Tan, Zhenrong Tang, Aishun Jin, Shengchun Liu

**Affiliations:** 1Department of Breast and Thyroid Surgery, The First Affiliated Hospital of Chongqing Medical University, Chongqing 400016, China; 2Department of Immunology, College of Basic Medicine, Chongqing Medical University, 1 Yixueyuan Road, Yuzhong District, Chongqing 400016, China; 3Chongqing Key Laboratory of Cancer Immunology Translational Medicine, Chongqing Medical University, 1 Yixueyuan Road, Yuzhong District, Chongqing 400016, China

**Keywords:** MED16, breast cancer, oestrogen receptor-positive, mTOR signalling pathway, autophagy

## Abstract

Recent studies have shown that the mediator complex (MED) plays a vital role in tumorigenesis and development, but the role of MED16 (mediator complex subunit 16) in breast cancer (BC) is not clear. Increasing evidence has shown that the mTOR pathway is important for tumour progression and therapy. In this study, we demonstrated that the mTOR signalling pathway is regulated by the expression level of MED16 in ER+ breast cancer. With the analysis of bioinformatics data and clinical specimens, we revealed an elevated expression of MED16 in luminal subtype tumours. We found that MED16 knockdown significantly inhibited cell proliferation and promoted G1 phase cell cycle arrest in ER+ BC cell lines. Downregulation of MED16 markedly reduced the sensitivity of ER+ BC cells to tamoxifen and increased the stemness and autophagy of ER+ BC cells. Bioinformatic analysis of similar genes to MED16 were mainly enriched in autophagy, endocrine therapy and mTOR signalling pathways, and the inhibition of mTOR-mediated autophagy restored sensitivity to tamoxifen by MED16 downregulation in ER+ BC cells. These results suggest an important role of MED16 in the regulation of tamoxifen sensitivity in ER+ BC cells, crosstalk between the mTOR signalling pathway-induced autophagy, and together, with the exploration of tamoxifen resistance, may indicate a new therapy option for endocrine therapy-resistant patients.

## 1. Introduction

Breast cancer (BC) is the leading cause of cancer-related death in women. According to GLOBOCAN 2020, the incidence and mortality rates of BC in women are 24.5% and 15.5%, respectively [1]. Early diagnosis and treatment of breast cancer is particularly important. Breast cancer is highly heterogeneous, so more and more attention has been paid to individualized treatment. It is very important to find new diagnostic markers and molecular targets for breast cancer treatment. Extensive research has demonstrated that the altered expression of signalling molecules can lead to breast cancer progression or treatment failure [2,3].

The treatment of BC varies for different subtypes. Currently, the biological subtypes of BC are classified on the basis of the expression of steroid receptors (oestrogen receptor (ER), progesterone receptor (PR) and human epidermal growth factor receptor 2 (HER-2)) by immunohistochemistry (IHC) [4]. Approximately 70% of BCs express oestrogen receptor α (ER), and ER-positive (ER+) tumours are predominantly of the luminal molecular subtype [5]. The first recommended therapy for ER+ BC is endocrine therapy [6]. Indications for endocrine therapy for breast cancer are estrogen receptor (ER) and/or progesterone receptor positivity. The level and source of estrogen in the human body before and after menopause is different, the choice of endocrine therapy drugs is also different, and the prognosis and treatment options for early and late tumors are also different. Tamoxifen is the first-line adjuvant therapy for premenopausal patients and the most commonly used endocrine drug. Approximately 30 to 40% of BC patients receiving tamoxifen adjuvant therapy still experience relapse or disease progression to a fatal advanced stage of metastasis within 15 years of follow-up [7]. The exact mechanism that leads to tamoxifen resistance during treatment with tamoxifen is still unclear. In recent years, there has been considerable evidence that autophagy-related factors, cell cycle modulators and a few transcription factors play important roles in tamoxifen resistance [8,9].

MED16, mediator complex subunit 16, which encodes a protein that is one of the constituents of the tail of the mediator complex (MED) [10], enables thyroid hormone receptor binding activity and transcription coactivator activity, and contributes to the positive regulation of transcription initiation from the RNA polymerase II promoter [11]. Previous studies have shown that MED plays a role in the expression of ER-regulated genes [12,13]. It is also known that MED mediates organ development, cell differentiation, mutation and alterations in the process of several diseases. Analysis of the mRNA expression of breast cancer clinical sample data from GEO (Gene Expression Omnibus) found that most MED subunits were differentially expressed in cancer and adjacent tissue. Therefore, we focused on MED16, which was reported in a previous study to be expressed at lower levels in thyroid cancer tissue than in adjacent normal tissue, and activated the TGFβ/SMAD signalling pathway [14]. However, whether the change in MED16 gene expression affects the biological function and therapy response of BC remains unclear. Through this research we indicated that MED16 overexpression activates the PI3K/AKT/mTOR signalling pathway in ER+ BC, which might be explored as a potential guide for clinical therapy.

## 2. Materials and Methods

### 2.1. Patients and Samples

Paired breast clinical human specimens (cancer and paracancerous tissue) originated from BC patients from the First Affiliated Hospital of Chongqing Medical University. All patients were diagnosed with BC by pathological biopsy and underwent BC surgery at the First Affiliated Hospital of Chongqing Medical University. The oestrogen receptor (ER) and progesterone receptor (PR) status of the patients were determined according to the results of IHC by the Clinical Diagnostic Pathology Centre of Chongqing Medical University. The study was approved by the Ethics Committee of Chongqing Medical University. All patients signed informed consent. 

### 2.2. Bioinformatics Data Analysis

Data showing MED16 in BC tissues and normal tissues were downloaded from The Cancer Genome Atlas Program (TCGA) (The Cancer Genome Atlas Program—National Cancer Institute) and the Gene Expression Omnibus database (https://cancerge-nome.nih.gov) (GEO42568). Data on MED16 expression in different subtypes of BC cells were obtained from the TCGA database and analysed with R (version 4.0.3, Auckland, New Zealand). Correlated genes of MED16 in BRCA were obtained from the UALCAN database (Ualcan.path.uab.edu/analysis). Patients without corresponding clinical information in the TCGA were excluded, and data for all remaining patients were analysed using the R package rms.

### 2.3. Cell Culture

Human ER-positive BC cells (MCF7 and T47D) and triple-negative BC cells (BT-549, MDA-MB-436, MDA-MB-231 and MDA-MB-468) were cultured in high-glucose DMEM with pyruvate (Gibco, Thermo Fisher Scientific, Waltham, MA, USA), 10% foetal bovine serum (FBS) (ExCell Bio) and 1% penicillin/streptomycin (Thermo Fisher Scientific, Waltham, MA, USA) at 5% CO_2_ and 37 °C in an incubator.

### 2.4. Plasmid Construction and shRNA and siRNA Transfection

All siRNA fragments were purchased from Tsingke Biological Technology. The primer target sequences of si#1, sh#1 and si#2, sh#2 were GAACTGCCTGGCTGTTGAA, CTTTCTCAACACGCCTGACAA. pLKO.1 was linearized with Age1 and EcoR1, and inserted into the shRNA fragment. The CRISPR guide sequences (KO#1 and KO#2 were GGCCATCACCTGCCTGGAGT, GGCCATCACCTGCCTGGAGT) were inserted into lentiCRISPRV2. To overexpress MED16, the MED16 CDS fragment was directly cloned into a lentiviral vector, pCDH-CMV-MCS-EF1a-CopGFP-T2A-Puro (System biosciences#CD513B-1). The forward primer was 5′-3′ CGTTTAGTGACCGTCATGCCACCATGTGTGATTTGCGGCGG, and the reverse primer was 5′-3′ ATAGTCATTGGTCTTAAAGGTAGTCAGGGGTAGCTGAGGGGC. Plasmid DNA transfections were performed using Lipofectamine 3000 (Thermo Fisher Scientific, Waltham, MA, USA) following standard protocols in accordance with the manufacturer’s guidelines.

### 2.5. Cell Proliferation and Drug Sensitivity Assay

Cells were plated in 96-well plates; when monitoring proliferation, approximately 3000 cells/100 µL per well were plated in a 96-well plate. Cell viability was measured with Cell Counting Kit-8 (CCK-8) reagent (Beyotime, Shanghai, China). After incubation for 2 h, the absorbance of the cells was measured at 450 nm. For the drug sensitivity experiment, after 5000 cells were seeded per well, they were incubated for 24 h and then treated with 4-OH tamoxifen for 72 h at different concentrations (0, 3, 6, 9, 12, 15 μM). Cell viability was measured as described before.

### 2.6. Colony Formation Assay

MCF7 and T47D ER+ BC cells (~800 cells/well) were seeded in 6-well plates. After 2 weeks of culturing, the cells were fixed with 4% paraformaldehyde for 15 min and then stained with crystal violet for 20 min at room temperature.

### 2.7. Immunohistochemistry (IHC) Analysis

The collected human tissue was fixed with a 4% formaldehyde buffer. The embedded tissue was then sliced into 4 μm thick slices. Tissue sections were incubated at 60 °C for 2 h prior to dewaxing and autoclaving at 115 °C for 3 min for antigen repair in citric acid buffer (pH 6.0), and endogenous peroxidase activity was quenched with 0.3% H_2_O_2_ solution for 15 min. Then, the slices were nonspecifically bound to normal goat serum blocking solution for 45 min and incubated with specific primary antibodies at 4 °C (dilution 1:100) overnight. Subsequently, the slices were treated with goat anti-rabbit secondary antibody for 30 min at room temperature. Protein expression was visualized using 3,3′-diaminobenzidine (DAB). Images were captured using a Leica microscope (Leica, Germany).

### 2.8. RNA Isolation, Reverse-Transcription Reaction and Quantitative Real-Time PCR (qPCR)

Total RNA was extracted from cultured cells with the Total RNA Extraction Kit (Tiangen, Beijing, China), and reverse transcription was performed with a 4xRT mix (MedChemExpress, Shanghai, China). Quantitative RT–PCR was performed in a 10-μL PCR mixture using SYBR Premix Ex TaqTM II (MedChemExpress) on a Bio-Rad CFX96 Real-Time PCR System (Bio-Rad Laboratories, Inc., Hercules, CA, USA). An initial cycle of 2 min at 95 °C, then 39 cycles at 95 °C for 30 s, a cycle of 30 s at 58 °C, and a cycle of 20 s at 72 °C. Three independent experiments were performed per group. Relative gene expression was normalized to GAPDH and assessed using the 2^−ΔΔCt^ method.

### 2.9. Western Blot Analysis

The following antibodies were used: anti-MED16 (THRAP5 polyclonal antibody) (Invitrogen, PA5-40517), anti-GAPDH (Proteintech, 60004-1-Ig), anti-c-MYC (Proteintech, 67447-1-Ig), anti-CCND1 (Proteintech, 60186-1-lg), anti-LC3A/B (CST, 12741), anti-P62 (Proteintech, 66184-1-Ig), anti-PI3K (BIMAKE, A5635), anti-phospho-PI3KP85α/γ/β (SOLAIBIO, K006379P), anti-AKT (Proteintech, 60203-2-Ig), anti-phospho-AKT (Ser473) (Proteintech, 66444-1-Ig), anti-mTOR (Proteintech, 66888-1-Ig), and anti-phospho-mTOR (Ser2248) (CST, 5536). The cells were collected on ice after two washes with prechilled PBS (4 °C). Protein lysates were prepared using RIPA lysis buffer (Beyotime, Shanghai, China) containing protease inhibitors (Beyotime, Shanghai, China) and phosphatase inhibitors (Beyotime, Shanghai, China), and the protein concentration was measured with a BCA Assay Kit (Beyotime, Shanghai, China). Extracted proteins (20 μg/10 μL/lane) from each group were separated by 10% sodium dodecyl sulfate–polyacrylamide gel electrophoresis (SDS–PAGE) (Bio-Rad Laboratories, Inc.). After blocking with fast blocking buffer (New Cell & Molecular Biotech, Suzhou, China) for 10 min at room temperature, the membrane was incubated with the specific primary antibody (dilution 1:1000) overnight at 4 °C. Then, after washing three times with Tris buffered saline containing Tween-20 (TBST), the membrane was incubated with the secondary antibody (dilution 1:1000) for 60 min at room temperature. Proteins were visualized by chemiluminescence using enhanced chemiluminescent substrates (Bio-Rad Laboratories, Inc.). Immunoreactive bands were developed using a chemical imaging system. GAPDH was used as a control.

### 2.10. 5-Ethynyl-2′-Deoxyuridine (Edu) Staining

The proliferation of BC cells was detected using the BeyoClickTM EdU Cell Proliferation Kit (C0075S, Beyotime, Shanghai, China) according to the manufacturer’s instructions. Briefly, 2 × 10^5^ cells were seeded in a 12-well plate for 24 h and incubated for 2 h at 37 °C with Edu working solution (10 μM) in the dark. After incubation, the cells were washed twice with PBS and fixed with 4% paraformaldehyde for 15 min. Next, the cells were permeabilized with 0.1% Triton X-100 for 15 min and washed three times with PBS. The cells were then incubated with DAPI (C0075S, Beyotime, Shanghai, China) for 5 min. Images were captured with a fluorescence microscope (Leica, Germany), where cells undergoing DNA replication during incubation showed red fluorescence, while nuclei were represented by blue fluorescence.

### 2.11. Cell Cycle Assay and Flow Cytometry Analysis

The culture supernatant was aspirated and rinsed once with PBS containing no calcium or magnesium ions. Pancreatic enzyme digestion was stopped after 2–5 min. The sample was transferred to a centrifuge tube, centrifuged at 1000 r/min for 5 min, and washed and centrifuged 1–2 times; the cell concentration was adjusted with the wash to 1 × 106/mL, and the sample was transferred to a BD FACS Verse Flow Cytometer for detection. The data were analysed using FlowJo software.

### 2.12. Mammosphere Formation Assay (MSF)

Breast cancer cells were seeded at 10,000 cells/mL in a 6-well ultralow attachment cell culture plate (Corning, Corning, NY, USA) and supplemented with 20 ng/mL bFGF (Peprotech, East Windsor, NJ, USA), 20 ng/mL EGF (Peprotech, East Windsor, NJ USA) and 2% B27 (Invitrogen, Waltham, MA, USA) in DMEM/F12 medium (Gibco, Thermo Fisher Scientific) that does not contain phenol red. Spheroid formation of the cells was observed daily for 10 days. On day 4, images were taken with a microscope (Leica, Wetzlar, Germany).

### 2.13. Statistical Analysis

All data were expressed as the mean ± SD of at least three independent experiments. GraphPad Prism 8.0 software (San Diego, CA, USA) was used for data analysis. Student’s *t*-test or one-way ANOVA was used to analyse differences between groups, and *p* * < 0.05 was considered statistically significant.

## 3. Results

### 3.1. MED16 Expression Is Markedly Upregulated in BC

We examined BRCA datasets from the TCGA and GEO databases to evaluate the differential expression of MED16 between BRCA and normal tissues. Analysis from TCGA BRCA datasets indicated that MED16 expression was higher in BC tissues (*n* = 1102) than in normal tissues (*n* = 113) (Figure 1A), and this was also shown in data from the GEO database (Figure 1B). The increased expression of MED16 in BC tissues was also demonstrated by the RT–qPCR of clinical specimens (Figure 1C) and (Table 1). Next, we measured the MED16 expression in different subgroups data from TCGA. As shown in Figure 1D, MED16 was upregulated in HR+ BC tissues, which was also shown in western blotting and the RT–qPCR of different cell lines (Figure 1E,F and Appendix A). We also detected the MED16 expression in ER+ BC tissues and normal paracancerous tissues by IHC, and MED16 was mainly expressed in the cytoplasm of cancer cells (Figure 1G).

### 3.2. Diagnostic and Survival Value Analysis of MED16

To determine the prognostic role of MED16 in BC patients, we analysed the relationship of MED16 expression with individual cancer stages, nodal metastasis status and patient age (Figure 2A). We downloaded the clinical information from TCGA-BRCA datasets. In this study, we used the R software package rms, integrated data on survival time, survival status, age, sex, stage, TNM and MED16 expression, and established nomograms using the Cox method to assess the prognostic significance of these features in 1023 samples (excluding the unknown information samples) (Figure 2B). We used the R software package MaxStat (maximally selected rank statistics with several *p* value parity version: 0.7–25). The optimal cut-off value of the risk score was calculated (Appendix A), with a minimum sample size greater than 25% and a maximum sample size less than 75%, and the optimal cut-off value was 0.429860830447361. Based on this, patients were divided into high and low groups, and the survival software package was further used. The survfit function of R was used to analyse the difference in prognosis between the two groups, the log-rank test was used to evaluate the significance of the difference in prognosis between different groups, and a significant difference in prognosis was finally observed (*p* = 0.0000000000000008) (Figure 2C). In conclusion, by calculating the risk coefficient of clinical characteristics related to MED16 expression, we divided the group into a high-risk group and a low-risk group and found that the group with a higher risk coefficient related to MED16 had a lower survival rate.

### 3.3. Knockdown of MED16 Inhibits ER+ BC Proliferation

To analyse the function of MED16 in BC, we used shRNA after validating the efficiency using the designed siRNA (Appendix A). The transfection of sh-MED16#1/2 into MCF7 and T47D cells was implemented to silence MED16 expression. RT–qPCR and western blotting showed that MED16 expression was markedly elevated and reduced in BC cells versus vector cells, respectively (Figure 3A,B and Appendix A). We also used CRISPR to construct MED16 knockout MCF7 cells (Appendix A). In addition to the function of MED16, we explored its regulatory mechanism in BC cells. CCK-8 assay showed that the proliferation of BC cells was decreased by the downregulation of MED16 (Figure 3C and Appendix A). The number of colonies was increased and decreased by MED16 upregulation and downregulation, respectively (Figure 3D). Moreover, Edu experiments confirmed that MED16 depletion decreased cell viability (Figure 3E). To further explore the effect of MED16 downregulation on BC growth, we examined the cell cycle effect of MED16 knockdown on MCF7 and T47D cells by flow cytometry, and the results showed that cell proliferation was slightly arrested in the G1 phase and that the number of cells in the S phase decreased during the experiment (Figure 3F). Moreover, c-MYC and CCND1 are markers of cancer cell proliferation and cell cycle progression, and western blotting showed that the expression of c-MYC and CCND1 was higher in the MED16 overexpression group (Figure 3G and Appendix A). In a word, MED16 promotes ER+ BC cell proliferation in vitro according to previous biological functional experiments.

### 3.4. Knockdown of MED16 Reduces Tamoxifen Sensitivity

To understand the relationship between MED16 expression and specific treatment in ER-positive BC, we investigated the role of MED16 gene expression and sensitivity to tamoxifen. Tamoxifen is an oral endocrine agent for early and premenopausal BC patients which has been commonly used for more than 30 years [15]. It competes with oestrogen for ERα and inhibits oestrogen’s stimulating effect on tumour growth and metastasis. We first explored the expression of the MED16 gene in the tamoxifen resistance database GSE125738 (Figure 4A). MED16 expression was remarkably lower in tamoxifen-resistant T47D cells (T47D-TAMR) than in wild-type cells. We generated 4OH-tamoxifen-resistant T47D cells and treated T47D and tamoxifen-resistant T47D cells with various doses of 4-hydroxytamoxifen (4-OHT), which is the active metabolite of tamoxifen, for 72 h to verify the efficacy of tamoxifen-resistant cells. CCK-8 assays detected cell viability and showed resistance to 4-OH tamoxifen (Figure 4B). Western blot analysis showed that MED16 expression was downregulated in T47D-TAMR cells (Figure 4C and Appendix A). As the time of 4-OHT exposure increased in MCF7 and T47D cells, the expression level of MED16 decreased (Figure 4D and Appendix A). We next investigated the role of MED16 in tamoxifen sensitivity. MED16 overexpression enhanced the inhibitory effects on cell viability in 4-OHT-treated MCF cells; conversely, MED16 downregulation in MCF7 cells reduced the sensitivity to 4-OHT (Figure 4E and Appendix A). The colony formation assay also demonstrated that the viability of MED16 knockout MCF7 cells was higher when the cells were treated with 1 µM 4-OHT (Figure 4F). It has become widely accepted that cancer stem cells (CSCs) play a vital role in cancer progression and in the resistance to chemotherapy [16]. To investigate whether MED16 targeting affects the stemness phenotype, we performed a mammosphere formation assay and found that MED16 knockout increased mammosphere growth compared to that in parental cells (Figure 4G). Based on these findings, we proved that low MED16 expression contributes to tamoxifen resistance.

### 3.5. Biological Functional Analysis of MED16-Related Genes

Since the function of MED16 in BC was confirmed, we investigated the pathways related to BC and molecular functions in different databases. First, we explored the genes positively and negatively correlated with MED16 in the UALCAN database (Appendix A). For the gene set functional enrichment analysis, we used the KEGG rest API to obtain the latest KEGG pathway gene annotation, which was used as the background to map genes into the background set. The R software package clusterProfiler (Version 3.14.3, Guangzhou, China), was used for enrichment analysis to obtain the results of gene set enrichment (Figure 5A,B). The minimum gene set size was set as 5, and the maximum gene set size was set as 5000. A *p* value of 0.01 and an FDR of 0.25 was considered to indicate statistical significance. In positive enrichment GO plots, the phospholipase D signalling pathway and endocytosis were significantly enriched, and autophagy, the mTOR signalling pathway, and endocrine resistance were also enriched. The cell cycle, PI3K/AKT and AMPK signalling pathways and endocrine resistance were enriched in genes negatively correlated with MED16 in the BRCA database.

### 3.6. Knockdown of MED16 Induces Autophagy by Inhibiting the PI3K/AKT/mTOR Signalling Pathway

The above functional enrichment analysis shows that autophagy and its related pathways are enriched in genes related to MED16. Autophagy is thought to induce resistance to antioestrogen treatment in BC cells [17]. Thus, to explore the effect of MED16 on autophagy, we first determined the expression levels of LC3 and P62 in the tamoxifen-resistant cells we constructed. The ratio of LC3B to LC3A protein expression level in T47D-TAMR cells was increased compared to that in wild-type cells, and P62 was decreased (Figure 6A and Appendix A). We also detected the protein levels of P62 and LC3 in the MED16 overexpression and knockdown groups, and western blotting experiments showed elevated autophagy levels after MED16 gene knockdown (Figure 6B and Appendix A). We also found that the ratio of the autophagy-related marker LC3B/A ratio decreased in tamoxifen-resistant T47D cells overexpressing MED16, demonstrating that MED16 affects the change in autophagy levels in BC cells (Appendix A). The functional enrichment analysis of genes related to MED16 showed that the PI3K/AKT/mTOR signalling pathway may be involved (Figure 5A,B), and our GSEA enrichment analysis also shows the enrichment of the mTOR pathway (Appendix A). Several studies have proven that the PI3K/AKT/mTOR pathway is a key regulatory pathway for cell growth and resistance to tumour treatment. Next, to verify whether autophagy is the cause of MED16 knockout resulting in reduced sensitivity to tamoxifen, we performed a recovery experiment with the autophagy inhibitor chloroquine (CQ). With the addition of CQ, autophagy was suppressed, reversing MED16 knockout caused by decreased sensitivity to tamoxifen (Figure 6C,D and Appendix A). We also explored the changes in the autophagy-related PI3K/AKT/mTOR signalling pathway after MED16 expression was changed. After MED16 was overexpressed in MCF7 cells, this signalling pathway was activated, while after MED16 expression was inhibited in MCF7 cells, the result was reversed (Figure 6E and Appendix A). We set out to explore whether an increase in drug resistance was caused by an increase in autophagy levels by inhibiting mTOR. MHY1485 is a potent cell-permeable mTOR activator that targets the ATP domain of mTOR. It inhibits autophagy by suppressing the fusion between autophagosomes and lysosomes. When MHY1485 was used, mTOR was activated, and the autophagy level decreased, thus reversing tamoxifen resistance caused by the decreased expression level of MED16 to a certain extent (Figure 6F). 

## 4. Discussion

BC is now the cancer with the highest incidence and mortality among women in the world, and details regarding its development and ideas for new treatments are urgently needed. Cancer progression and even death due to tamoxifen resistance have become major problems in improving the survival rate of patients with ER+ BC [6]. The exact mechanism that leads to tamoxifen resistance during treatment with tamoxifen is still unclear. In recent years, there has been considerable evidence that autophagy-related factors, cell cycle modulators and a few transcription factors play important roles in tamoxifen resistance [18,19]. 

In this research, the expression level of MED16 in breast cancer and the role for endocrine therapy were detected for the first time. Previous studies have shown that the mediator complex (MED) plays a role in the expression of ER-regulated genes [12,19,20]. It is also known that MED mediates organ development, cell differentiation, mutation and alterations in the process of several diseases. MED16, mediator complex subunit 16, which encodes a protein that is one of the constituents of the tail of MED [10], enables thyroid hormone receptor binding activity and transcription coactivator activity and contributes to the positive regulation of transcription initiation from the RNA polymerase II promoter [11]. We found that MED16 expression is elevated in breast cancer tissues as well as in cells, and knocking down its expression inhibits the proliferation of ER+ breast cancer cells.

Additionally, we explored the effect of MED16 on the sensitivity of tamoxifen, the most common endocrine treatment. We observed that the overexpression of MED16 increases sensitivity to tamoxifen in ER+ breast cancer and vice versa, we also found that knockdown of MED16 promotes autophagy levels. Autophagy is a process of using lysosomes to dissolve their own cytoplasmic proteins and damaged organelles in the regulatory decline of autophagy-related genes [21,22]. On the one hand, autophagy can accelerate the death process through self-phagocytosis; on the other hand, autophagy can protect the survival of stressed and damaged cells by delaying their apoptosis [23,24]. Recent studies have examined whether tamoxifen-resistant BC cells have a greater degree of autophagy than sensitive cells [25,26]; the autophagy-correlated signalling pathway PI3K/AKT/mTOR contributes to tamoxifen resistance [27], and MAPK/ERK pathway activation has also been demonstrated to be conducive to tamoxifen resistance [28,29].

We further performed an enrichment analysis of similar genes to MED16 and found that MED16 was associated with the PI3K/AKT/mTOR signalling pathway. Our findings demonstrated that silencing MED16 could significantly decrease the protein level of markers of the PI3K/AKT/mTOR signalling pathway. The PI3K/AKT signalling pathway is overactivated in various tumours and strongly correlated with cell progression, the cell cycle, apoptosis, and autophagy [30]. Previous studies have demonstrated that inhibition of the PI3K/AKT signalling pathway is crucial for cell cycle regulation, which can decrease the phosphorylation of AKT and lead to higher p21 expression, which can in turn lead to G1 phase cell cycle arrest [31]. The mTOR signalling-related pathway also plays a vital role in tumour occurrence and progression. It has been demonstrated that activation of autophagy reduces the therapeutic effect of radiotherapy and chemotherapy on BC [32]. Previous studies also proved that the activation of mTOR signalling inhibits the autophagy-inducing capacity of autophagy-related protein 1 (Atg-1) [33], which is a node in several different signalling pathways which regulate autophagy. This result is consistent with the observation that by inhibiting autophagy and reversing the reduced sensitivity of tamoxifen caused by the downregulation of MED16, we propose that autophagy inhibitors, in combination with tamoxifen, may lead to a better prognosis for patients with ER+ breast cancer.

## 5. Conclusions

In summary, we found that MED16 plays an important role in breast cancer progression and treatment, and that its mechanism mainly depends on the activation of mTOR. PI3K/AKT/mTOR inhibition caused by MED16 knockdown may lead to a slower proliferation of breast cancer cells and autophagy enhancement, which leads to a decreased sensitivity to tamoxifen. For ER+ BC patients with low MED16 expression, it is recommended that tamoxifen be used in combination with autophagy inhibitors to achieve better results. In the future, the specific activation mechanism of MED16 on the mTOR signalling pathway may be explored.

## Figures and Tables

**Figure 1 life-12-01461-f001:**
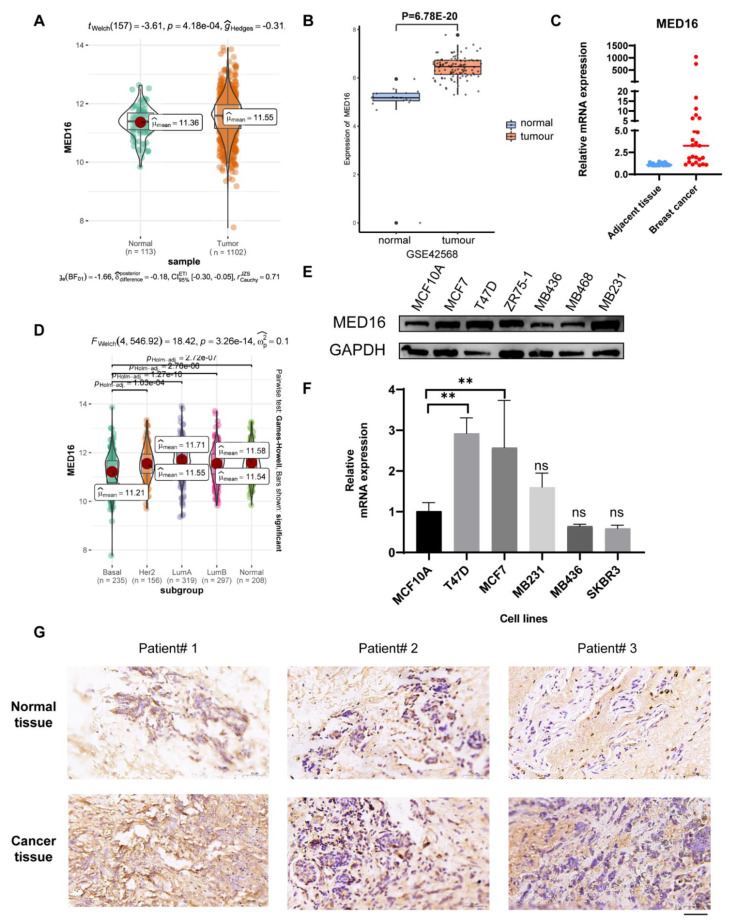
The expression of MED16 is associated with breast cancer tumorigenesis. (**A**,**B**) Gene expression data of MED16 in breast cancer and normal tissues from the TCGA and GEO databases showed that MED16 mRNA level was overexpressed in human breast cancer (**C**) RT-PCR was adopted to detect the mRNA expression level of MED16 in tumour and adjacent tissues (n = 25). (The values are compared to normal group and are represented as the means ± S.E.M (ns: not significant, ** *p* < 0.01). (**D**) MED16 expression in breast cancer subtypes from the TCGA cohort. (**E**,**F**) Western blot and RT-qPCR were used to detect the MED16 expression in different subtype BC cell lines. (**G**) IHC staining of MED16 expression in normal and ER+ BC tissues (the bar on the figure represents 50 µm), mostly located in the cytoplasm.

**Figure 2 life-12-01461-f002:**
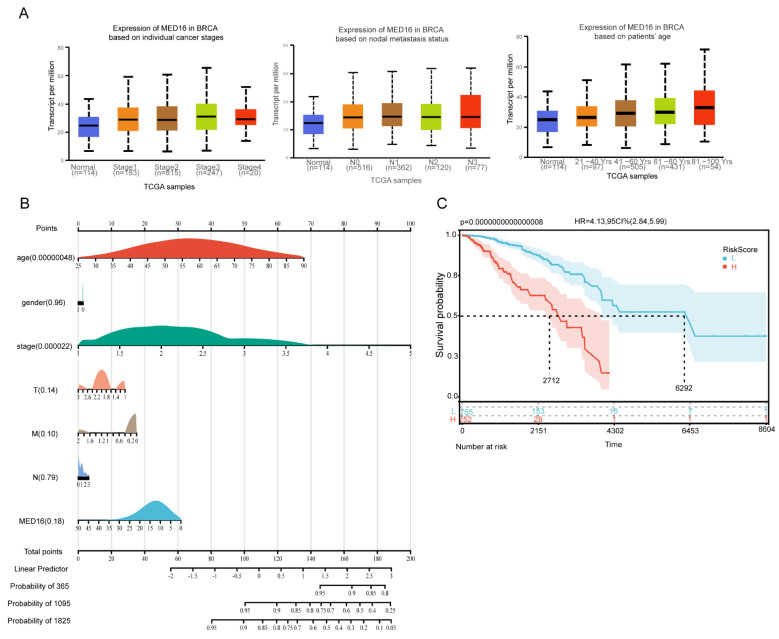
The prognostic value of MED16 in BC patients in different databases. (**A**) Gene expression data of MED16 in different clinical features in TCGA databases from UALCAN. (**B**) Nomograms were established using the Cox method, and the prognostic significance of these features in 1023 samples was assessed. (**C**) Log-rank test method was used to evaluate the significance of the prognostic differences between different groups of samples with Risk score.

**Figure 3 life-12-01461-f003:**
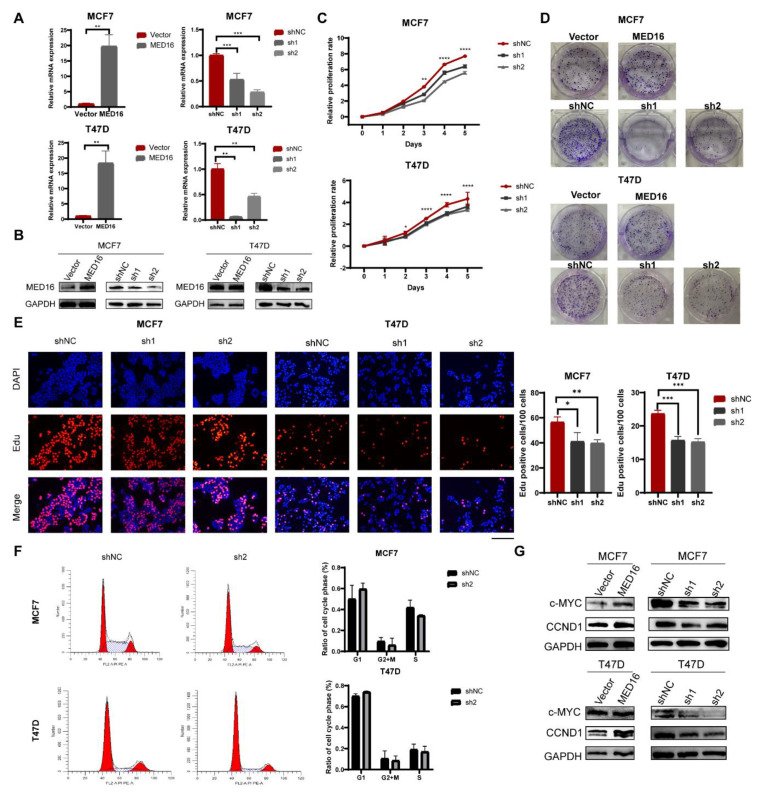
Knockdown of MED16 hinders proliferation of BC cells. (**A**,**B**) RT−qPCR and western blot revealed the expression of MED16 was overexpressed and inhibited remarkably in MCF7 and T47D. (**C**) Growth curve of MCF7 and T47D cells infection of MED16−negative control and MED16−knockdown was displayed. Relative proliferation ratio against day 0 were observed for 5 days (* *p* < 0.05; ** *p* < 0.01; *** *p* < 0.001; **** *p* < 0.0001). (**D**) Colony formation of MED16−overexpression cells and MED16-knockdown cells compared with MED16-negative control cells. (**E**) Representative fluorescence images of Edu staining of MCF7 and T47D cells. Proliferating MCF7 and T47D positively stained with Edu showed red colour. Cell nuclei stained with DAPI showed blue colour. The length of bar represents 750 μm, chart of proliferating cells (red) to total cells (blue). (**F**) Knockdown of MED16 induced G1−phase cell cycle arrested. Flow cytometric assay was employed to analyse the cell cycle distribution. Percentage of cells in the G1 phase was evidently increased, compared with the control group. (**G**) c-MYC, cyclinD1 protein expression in MCF7 and T47D cells with control (Vector) and MED16-overexpressed cells, control (shNC) and MED16−knockdown cells (sh1, sh2).

**Figure 4 life-12-01461-f004:**
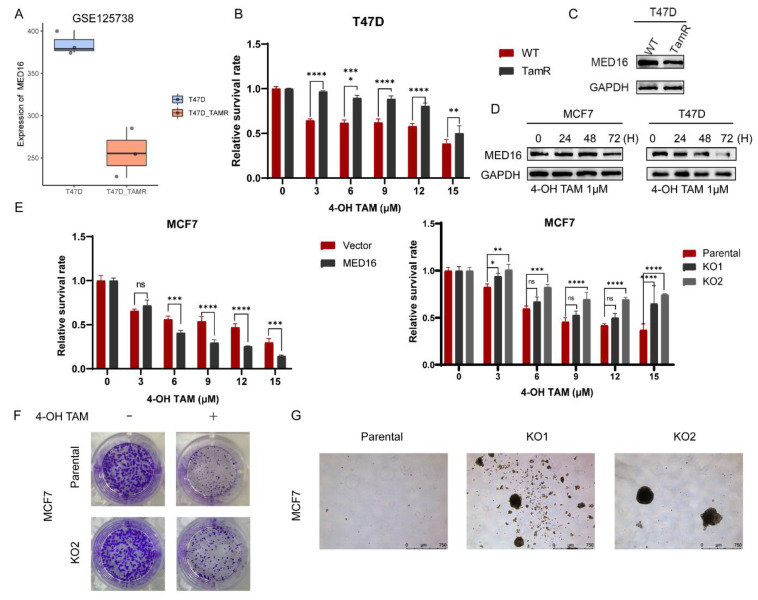
Downregulation of MED16 reduces tamoxifen sensitivity in ER+ breast cancer cells. (**A**) Gene expression data of MED16 in T47D wild-type and tamoxifen-resistant T47D from GEO database (GSE125738). (**B**) 4OH-Tamoxifen sensitivity of tamoxifen-resistant T47D. (ns: not significant, * *p* < 0.05; ** *p* < 0.01; *** *p* < 0.001; **** *p* < 0.0001) (**C**) Protein level of MED16 in WT and TAMR of T47D. (**D**) The MED16 protein expression levels of different time gradients at 1 μM 4OHT treatment. (**E**) Effect on control (vector) and MED16-OE, control (Parental) and MED16 knockout (KO1, KO2) MCF7 cells with different doses of 4OH-TAM at 72 h respectively. (**F**) Colony formation assay of untreated and treated with 4OHT (1 μM)in control (Parental) and MED16 knockout (KO2) MCF7 cells. (**G**) A micrograph showing mammosphere formation in control (Parental) and MED16 knockout (KO2) MCF7 cells. The length of bar represents 750 μm.

**Figure 5 life-12-01461-f005:**
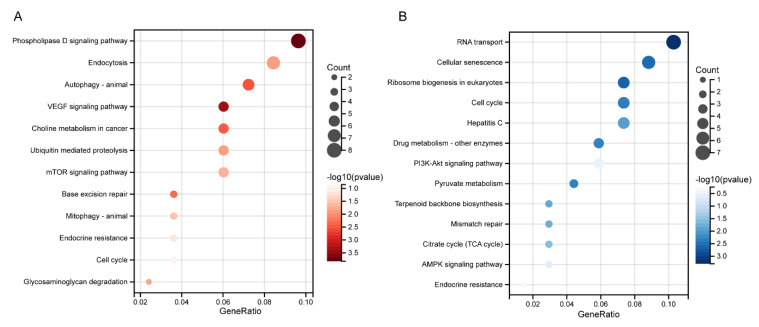
Functional annotation of MED16-related genes. (**A**) KEGG plot of positively related genes of MED16, indicated in red. (**B**) KEGG plot of negatively related genes of MED16.

**Figure 6 life-12-01461-f006:**
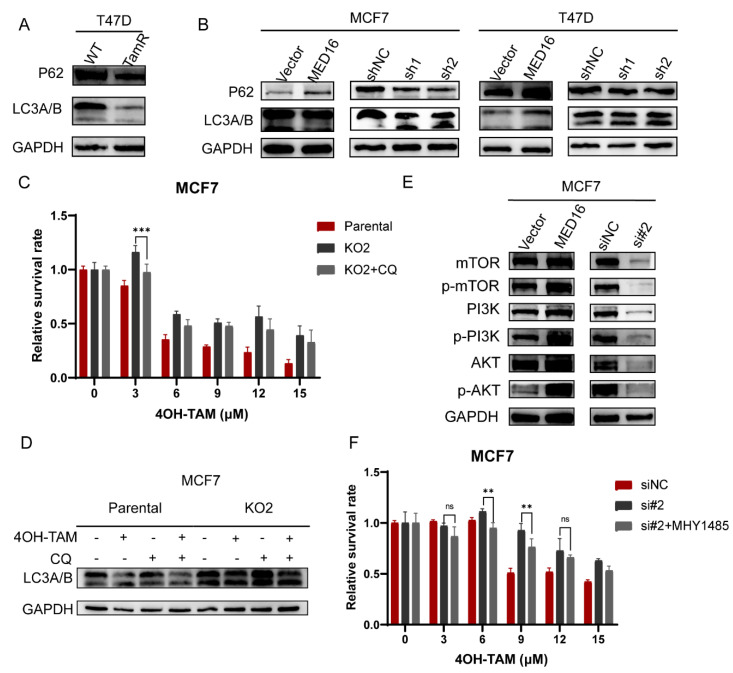
Knockdown of MED16-induced autophagy and inhibited PI3K/AKT/mTOR pathway in ER+ BC cells. (**A**) Autophagy marker P62, LC3A/B protein expression in T47D-WT and TamR-T47D. (**B**) Autophagy-related marker P62, LC3A/B protein expression in overexpression of MED16 and knockdown of MED16 in MCF7 and T47D. (**C**) The effect on control (Parental), MED16 knockout (KO2) and autophagy inhibitor CQ treated (KO2 + CQ) MCF7 cells with different doses of 4OH-TAM at 72 h. (ns: not significant, ** *p* < 0.01; *** *p* < 0.001) (**D**) Autophagy marker LC3A/B protein expression in 4OH-TAM and CQ treated in control (Parental), MED16 knockout (KO2) MCF7 cells. (**E**) mTOR, phospho-mTOR, PI3K, phospho-PI3K, AKT, phospho-AKT protein expression in overexpression of MED16 and knockdown of MED16 compared with control MCF7 cells. (**F**) Effect on control (Parental), MED16 knockout (KO2) and mTOR activator-treated (KO2 + MHY1485) MCF7 cells with different doses of 4OH-TAM at 72 h.

**Table 1 life-12-01461-t001:** Patients characteristics: ER estrogen receptor, PR progesterone receptor, HER2 human epidermal growth factor receptor 2.

Characteristics	HR Positive (N = 18)	HR Negative (N = 7)	Total (N = 25)	*p* Value	FDR
Age (years)				0.01	0.09
>50	6	7	13		
≤50	12	0	12		
Tumour size (cm)				0.79	1
>2	10	5	15		
≤2	8	2	10		
Histological grade				0.52	1
II	12	3	15		
III	6	4	10		
Lymph node metastasis				0.37	1
Negative	5	4	9		
Positive	10	2	12		
Unknown	3	1	4		
ER status				0.000047	0.00047
Negative	1	7	8		
Positive	17	0	17		
PR status				0.01	0.09
Negative	6	7	13		
Positive	12	0	12		
HER2 status				0.34	1
Negative	10	6	16		
Positive	8	1	9		
Ki-67 (%)				0.56	1
>30	5	3	8		
Positive	2	0	2		
≤30	11	4	15		
Molecular subtypes				0.00005	0.00047
HER2 positive	0	1	1		
Luminal A	10	0	10		
Luminal B	8	0	8		
Triple Negative	0	6	6		
Adjuvant therapy				0.31	1
No	11	2	13		
Yes	7	5	12		

## Data Availability

The datasets generated during and/or analysed during the current study are available from the corresponding author upon reasonable request.

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
