# Peer review of "MED16 Promotes Tumour Progression and Tamoxifen Sensitivity by Modulating Autophagy through the mTOR Signalling Pathway in ER-Positive Breast Cancer"

_life, 2022, doi:10.3390/life12101461_

Round 1

Reviewer 1 Report

Review for “MED16 promotes tumour progression and tamoxifen sensitivity by modulating autophagy through mTOR signalling pathway in ER positive breast cancer”

Study is cleverly designed and continues the previous research of this group. Manuscript is well-written and easy to follow for even the researchers outside of this field of study. Introduction provides enough background and needs no additional references. Research is appropriate and well-designed, methods are adequate and in line with the latest research trends with the results clearly presented. However, some of the conclusions do not seem to be, or are insufficiently, based on the presented results.

Furthermore, I would suggest in all Wb Figures to quantify the protein band density and add graphs of data normalized to GAPDH as it is easier to see the differences in protein expression in this way. Also, please pay attention to not use terms "knockout" and "knockdown" interchangeably (i.e. paragraph 3.3) as they are not synonyms. Check this through the manuscript and correct where applicable.

Authors should be careful about stating the "evident increase of cells in G1 phase" as that data is not significantly different compared to the control cells, when you take into account the error bar size. Furthermore, what is the reason sh1 was not included in cell cycle experiments?

Lastly, Authors should provide more data/explanations on the theory that the suppression of autophagy reverses the MED16-linked decrease in sensitivity to tamoxifen, as the differences in Fig. 6C/D are small and not statistically significant. Also, Discussion is not long and mainly lists the already known data, and not going into detail on the mechanisms behind reported results.

The following is the list of some additional issues that need to be addressed before the manuscript is, in my opinion, suitable for publication.

Line 4: Please change to "ER-positive".

Line 38: Please change to "extensive research has demonstrated".

Line 60: Too long of a sentence. I suggest splitting it and rewriting as: "Analysis of mRNA expression of breast cancer clinical sample data from GEO found that most MED subunits were differentially expressed in cancer and adjacent tissue. Therefore, we focused on MED16, which was reported in a previous study that MED16 was expressed at lower levels in thyroid cancer tissue than in adjacent normal tissue and activated the TGFβ/SMAD signalling pathway."

Line 61: Please explain the abbreviation “GEO” (Gene Expression Omnibus).

Line 67: Please change to "indicated that MED16 overexpression activates PI3K/ AKT/ mTOR signalling pathway".

Line 76: Abbreviation IHC already explained in the Introduction section. I suggest removing the explanation here and using the abbreviated form in further text. Same suggestion for the rest of the abbreviations.

Line 117: Why were only these breast cancer cell lines tested for colony formation?

Line 127: Please write the dilution for the secondary antibody used.

Line 129: Please add full information about the manufacturer Leica. Same suggestion for other manufacturer mentions (i.e. Tiangen).

Line 153: Please write the dilutions for primary and secondary antibodies used in Western blot experiments.

Line 162: In cell concentration write 5 in superscript font.

Lines 165-168: Please change verb tense to past tense. “Cells were”, Images were”, showed red fluorescence”, “nuclei were represented”.

Line 171: Please change to "was stopped after 2-5 min".

Line 177: Different number format than the one used in paragraph 2.5. Please correct this and use consistent form throughout the manuscript.

Line 184: Please change to "data were expressed".

Line 201-202: Please correct words to “characteristics" and “receptor2”.

Table 1: Please change “Tumor size” to “Tumour size” as this form is dominantly used in the manuscript.

Line 211: Could you explain in more detail what is exactly located mostly in the membrane as in previous paragraph it says MED16 is located mostly in cytoplasm. Please clarify this.

Line 240: Could Authors elaborate why they picked these two cell lines for this experiment?

Line 245: Could you clarify this sentence, the part about upregulating the MED16 decreases the proliferation? Fig. 3C and S1C only show downregulated MED16 via shRNA and siRNA.

Line 248: Use either "Edu" or "EdU" term through whole manuscript including the Figures.

Line 258: Unclear sentence, word "expression" used twice in same sentence, how come mRNA expression can be measured by Wb? Please write this in clearer form.

Line 262: Please change to "MED16-overexpressed cells and MED16-knockdown cells compared with".

Line 287: Please specify the concentration of 4-OHT used in colony formation experiments.

Line 297: Please correct to "tamoxifen".

Line 329: “The protein expression level of LC3 in T47D-TAMR cells was increased compared to that in wild-type cells”. Could you elaborate on this as this is not clear in Wb data in Fig. 6A? WT samples appear to have more LC3A/B protein expression.

Reviewer 2 Report

Please consider to update the introduction as example -please explain more the indication for hormonotherapy in breast cancer ER positive as mention in line 46

Reviewer 3 Report

This study investigated the key role and mechanism of MED16 in the tumorigenesis of luminal type breast cancer. Results from clinical big data, histological analysis, and non-clinical studies were well presented. Please consider the following comments.

1. Statistical analysis in Figure 1c is missing. And, specific information on the molecular subtypes of the patients in Figure 1G should be provided.

2. the demographic and etiological information provided in Figure 2B should be provided as a separate table.

3. Discussion of the specific mechanism of MED 16 acting on inhibition of mTOR is scarce. This information should be clearly discussed.
